# Transitional Justice Process and the Justice Theory of Roland Dworkin

**Helen Gyr**

Independent Researcher, 3012 Bern, Switzerland; helen.gyr@denkbilder.ch

**Abstract:** The determination of *truth* in the aftermath of war aiming at establishing *justice* and *peace* is a key element of a transitional justice (TJ) process. The theory of justice of Roland Dworkin deals with an approach in which the interpretation of values such as *equality, liberty* or *truth* are paramount. Dworkin's theory of justice is applied to constitutional states and lays out how democratic values are negotiated. The goal of a TJ process is to lead a state towards democracy after a war or internal armed conflict. TJ processes as well as Dworkin's theory of justice are to be understood as dynamic, which implies that they are subject to constant change and thus to be considered in their respective social, cultural, political, and economic contexts. This paper explores the relationship between *truth* and *justice* in the framework of a TJ trial and Roland Dworkin's theory of justice. The TJ process in Colombia serves as a case study because that was where I conducted field research in TJ in 2019.

**Keywords:** transitional justice; truth; justice; theory of justice by Roland Dworkin; interpretive approach; democracy; legal understandings; peacebuilding; Colombia

## 1. Introduction

The question of how *truth* is determined after a war in order to establish *justice* cannot be answered conclusively and can be approached from different scientific perspectives. After a war, destroyed places must be rebuilt on the one hand; on the other, social cohesion has to be restored. The establishment of peace thus requires a multi-layered reappraisal on different levels, as the question of who belongs to society and how political power is distributed are often at the root of violent and armed conflict.

After a violent conflict, which can either be between states or internal, the relationship between government and society is often disrupted because the state is weakened or dysfunctional. This can be due the fact that a government no longer has the support of a majority of the population, as was the case in Colombia, for example.

The determination of truth has been an overall goal in TJ processes around the world. At the same time, TJ processes must be understood within the respective zeitgeist and specific conflict situation, which in turn can lead to highly divergent results. TJ processes have become increasingly popular in Latin America, especially after the Cold War, and have led to different outcomes (cf. Encarnación 2022; cf. Gonzales Ocantos 2020; cf. Teitel 2003, 2014).[1] In this article, I shall be referring to the transitional justice process in Colombia as a case study for transitional justice (TJ).[2]

In the turbulence of a so-called post-war period,[3] TJ as an international instrument offers a variety of approaches to problem solving so as to (re)establish democracy in a state. The goal of a TJ process is to determine truth in order to establish *justice* and guarantee stable

---

1 The best-known exampels are Argentina and Brazil, which deal with their past military dictatorships in very different ways.
2 In 2019, I was in Colombia for three months conducting field research on the topic "Coming to terms with sexual violence during the internal armed conflict in Colombia." The field research is a basis for a documentary film "la verdad no es una prostituta" which is still in development (accessed on 30 March 2023).
3 By post-war period I mean any period after a war, so not just the post-war period after WW2.

peace (Teitel 2003; Werle and Vormann 2018). A TJ process can feature different mechanisms, such as a special court or a truth commission, that are used in the process during a certain period of time to achieve the goals set (cf. Elcheroth and De Mel 2022, p. 1 f.; cf. Werle and Vormann 2018, p. 6).

Roland Dworkin's (2011, 2014) theory of interpretation and justice revolves around the concept of a successful way of life within a liberal constitutional state. Essentially, it is about how a society can implement *equality, liberty,* and *law* in a democratic state. Law and morality are to be understood interpretively and as interrelated elements (Dworkin 2014, pp. 22f., 678 f.; cf. Ibric 2022, p. 139). Dworkin's interpretive and justice theory provides an interesting basis for examining the relationship between *liberty* and *justice* in relation to *truth-finding* within a TJ process.

TJ seeks to facilitate elementary coexistence of a society within a state, by guaranteeing *liberty, equality,* and *justice* through *law* (cf. Dworkin 2014, pp. 15–21). In this context, the interplay of law and state must be clarified, as both are supposed to guarantee peaceful coexistence in society by means of legal structures. In every postwar period, *justice* is a concrete demand; however, there cannot be *justice* without *truth* (cf. Dworkin 2014, pp. 15–21, 274, 307, 319). The question is how *truth* can be determined within a TJ process in such a way that *justice* can be established. The demand for *justice* is universal; nonetheless, the way in which it can be negotiated, in the context of different conflict situations and conceptions of *justice* in relation to the establishment of *truth,* must be clarified on a case-by-case basis.

The concepts of "liberty", "truth", and "justice" are presented in this paper in the context of value attributions and interpretations. Their definition is not to be regarded as conclusive, but rather as a framework for their understanding and discussion in a society or TJ process.

In this paper, the theoretical concept of TJ is first briefly introduced; thereafter, the TJ process and mechanisms in Colombia are explained in more detail. Subsequently, the concepts of *equality, liberty, democracy, justice,* and *truth* according to Dworkin's theory of justice will be presented and placed in the context of the TJ process in Colombia.

## 2. Transitional Justice (TJ)

*2.1. Transitional Justice: A Theoretical Concept in Transition*

As a dynamic process, TJ is subject to constant change and must be considered in its respective historical context (Teitel 2003). The concept of TJ has gained popularity since the 1990s, particularly with the publication of Neil Kritz's homonymous paper. The central aspect of Kritz's definition is the specification of the process as a transitional phase from a previously *dictatorial* to a *democratic* state (Werle and Vormann 2018, p. 3; cf. Gyr 2020, p. 155). TJ has since often been understood as a *mechanism* or *toolbox* that provides a concerned state with options on how to *truthfully* address past crimes in order to guarantee stable future peace (Elcheroth and De Mel 2022, p. 1 f.; Werle and Vormann 2018, p. 6). Such mechanisms of TJ include special courts, truth commissions, or reparations.

However, calling TJ a toolbox certainly falls short. Jens Ohlin for example, analyzes the two terms *transition* and *justice* separately and in their relationship to each other. According to Ohlin, *justice* is a term oriented towards normative and universal guidelines in the context of finding justice. Moral values are central to this concept. The term *transition,* on the other hand, places itself in the context of a political state of emergency, says Ohlin (2007, p. 51). There is a strained connection between the philosophical idea of *justice*—with its universal character—and the *transition* of a society—in a political state of emergency (cf. Gyr 2020, p. 156). Their fractured relationship is also detectable in the link between freedom and justice in the context of finding truth within a TJ process. After a war, justice is a concrete demand that cannot be achieved without truth (see introduction; cf. Dworkin 2014, pp. 15–21, 274, 307, 319). However, the exact shape of truth-finding in the aftermath of an armed conflict cannot be determined conclusively in a TJ process, because it is contingent on the political situations and actors involved in each individual case. Both are shifting

and potentially conflicting elements after a war as different actors have diverging interests, especially when it comes to negotiating liberty and equality within a state.

TJ according to Kritz—depicted as a process from a *dictatorial* to a *democratic* state—is a phase of transition. In the limited time span of a TJ used after violent conflict and massive human rights violations, Ruti G. Teitel (2003, p. 69 f.) argues that both the historical context and the means by which truth is determined to establish justice are important (cf. Gyr 2020, p. 157). After all, the goal is to establish justice and bring about democratic change. In her historicization of TJ, Teitel (2003, pp. 69–94) divides TJ into the following phases: *post-war TJ, post-Cold War TJ*, and *steady-state TJ*.

Teitel (2003, p. 90) understands *post-war TJ* as the influence of World War I and World War II on the concept of TJ. The Nuremberg and Tokyo Tribunals are examples of this, which, through their collective sanctions and international (military) jurisdiction, shaped this phase significantly. The replacement of national jurisdiction with international jurisdiction and international policy are key factors in this phase. They give TJ an unrestricted and universal character (Teitel 2003, p. 72; cf. Gyr 2020, p. 157).

According to Teitel (2003), the unrestricted and universal character of TJ is not questioned until the *post-Cold War* phase. Teitel (2003, pp. 75, 78) considers the *post-Cold War* period as the phase ushered in by the collapse of the Soviet Union, the beginning of which is related to the liberation movements in South America in the late 1970s and which ends with the Soviet Union's dissolution in 1991. Moreover, it can be demonstrated that new political beginnings after war have been increasingly linked to TJ issues. The International Criminal Tribunal for Rwanda (ICTR) or the Truth and Reconciliation Commission (TRC) of South Africa exemplify this phase (Anders and Zenker 2014, p. 395). The restorative model, in which the aim is to come to terms with past crimes and reconstruct history, has been gaining in importance. Increasingly, truth commissions have been used for historical assessment. Truth commissions thereby represent national reconciliation and special courts embody the establishment of justice. Together, they aim at guaranteeing peace (Teitel 2003, p. 77 f.) According to Richard A. Wilson (2003, p. 369), the truth commission can be understood as a constitutional and social approach to the reconciliation of conflicting parties (cf. Gyr 2020, p. 158).

Therefore, TJ consists in the attempt to respond to the different conflicts in a situational manner and to cooperate with the state concerned to the extent necessary to achieve the goal of peace and reconciliation (Teitel 2003, p. 77 f.).

In the establishment of peace and reconciliation, organizations from outside the state structure are becoming increasingly important, such as non-governmental organizations or churches (Teitel 2003, p. 83 f.; cf. Elcheroth and De Mel 2022, p. 10 f.). However, the growth of non-governmental actors leads to complex relationships between state institutions, international organizations, and local groups (Anders and Zenker 2014, p. 396 f.).

In the first phase of *post-war TJ*, the conventional and legal process played an important role in determining winners and losers. In *post-Cold War TJ*, the focus is on reconciliation resulting in a new beginning as a united state (Teitel 2003, p. 83 f.).

The third phase, *steady-state TJ*, is based on standardized and normalized procedures to create a liberal and democratic state under the rule of law from an "unjust" state after an internal armed conflict. In this phase, the general application of human rights gradually replaces the phase of contextual, local, and limited negotiation (Teitel 2003, p. 89 f.).

In this context, the relationship between freedom and justice is addressed with regards to the establishment of truth within a TJ process. Here, the question that arises is what is meant by truth in a TJ process and how does it relate to liberty and justice, especially if the TJ process is to be understood as a dynamic process.

*2.2. Transitional Justice in Colombia: Comprehensive System of Truth, Justice, Reparation and Non-Repetition—SIVJRNR*

After years of conflict and failed negotiations between the government and the guerrilla group *Fuerza Armada Revolucionaria de Colombia* (FARC), the Colombian government

and the FARC held new negotiations in October 2012 under international observation. Four years later, a peace treaty was negotiated, but narrowly rejected by the Colombian people in a referendum on 2 October 2016. A new peace treaty was then negotiated on 30 November 2016; however, it was not put to the vote (Werle and Vormann 2018, p. 287).

The negotiated peace agreement stated that TJ would be established in Colombia. The legal basis for the specific design of TJ in Colombia and its mechanisms was laid on 4 April 2017 (SIVJRNR). The first article introduces the concept of *Sistema Integral de Verdad, Justicia, Reparación y No Repetición* (SIVJRNR), which aims to regulate the termination of the armed conflict and ensure the establishment of stability and the permanence of peace.

The following mechanisms and measures are included in the SIVJRNR: a Truth Commission (CEV), a Special Unit for the Search of Missing Persons (UBPD), a Special Court (JEP), and measures for integral reparations. Together, they are intended to prevent a recurrence of conflicts and violent clashes in the future (JEP).

*Truth* and *justice* are explicitly mentioned in the name of the TJ of Colombia (SIVJRNR) and are institutionalized through the Truth Commission CEV and the Special Court JEP. Due to the temporary nature of the TJ process, the mandates of the aforementioned institutions are limited in time. Overall, all mandates of the SIVJRNR aim at determining *the truth* and establishing *justice*; therefore, the results of the different mandates must be considered jointly. The Colombian TJ nomenclature, however, provides no indication of the role and importance that *liberty* should take within the TJ process. The fact that the first peace treaty was (narrowly) rejected by the Colombian people in a referendum and the second one was not brought before the people shows the people's discord over the peace treaty and the associated TJ process. The disagreement also became evident in the presidential elections. In 2018, Ivan Duque was elected as a candidate who clearly positioned himself against the planned implementation of the peace treaty of his predecessor Juan Manuel Santos (cf. García Pinzón 2020, p. 2). In 2022, Gustavo Petro was the first member of a left-wing party to win the presidential elections. He was also a former member of the guerrilla group *Movimiento 19 de abril*. The TJ processes are always shaped and influenced by the political situation in a country, which, as described by Ohlin (2007, p. 51) points to the tension between *transition* and *justice* (see Section 2.1).

2.2.1. Truth Commission CEV

*"Hay futuro si hay verdad"* (There is a future if there is truth) is the official slogan of the Colombian Truth Commission, which has been tasked with the historical reappraisal of the internal armed conflict that has lasted more than fifty years. The mission of the Truth Commission is described as follows in its constitutional act dated 4 April 2017: "The Commission is an independent and temporary body with the specific task of providing clarification regarding events during the armed conflict. It aims to provide clarity regarding the complex interconnections in society so that victims and responsible parties can be identified. Furthermore, violations of human rights during the armed conflict should be clarified and the indirect as well as the direct involvement of actors should be shown. In this respect, the individual and collective responsibility of those involved should be made visible. However, the truth commission has no legal status. Its purpose is to promote peaceful coexistence in society by establishing the truth and thereby preventing the recurrence of violent conflicts" (JEP, Actos legislativos No. 01 del 4 de Abril de 2017).

The Truth Commission (CEV) commenced its work in November 2018. Its mandate was limited to three years. It collected testimonies from victims, created public and private archives, documented human rights violations, identified victims and those responsible, and presented proposals that were included as recommendations in the final report (CEV).[4]

On 28 June 2022, the Truth Commission presented its final report in Bogotá. The final report comprises ten chapters, documenting serious human rights violations in ten volumes

---

[4] Observation protocol on 7.3.19 for the occasion: Evento de lanzamiento de la cartilla "Participación de las víctimas en el Sistema Integral de Verdad, Justicia, Reparación y No Repetición"—OACNUDH.

totaling six thousand pages. The report is not legally binding and makes recommendations for the incumbent government as well as for the population. The central demand of the Truth Commission is that the government should consistently enforce the peace treaty and become active above all in the rural regions of Colombia, where there are still armed groups (cf. CEV, Hallazgo y Recomendaciones).

The slogan of the Colombian Truth Commission, *"Hay futuro si hay verdad,"* promises a future if there is truth. However, what will happen if the truth presented by the CEV is not accepted remains open. It is unclear what consequences the human rights violations documented by the Truth Commission will have. One of the goals of the Truth Commission is to ensure that there is no recurrence of violent conflict. Since violence in Colombia has once again increased after the peace treaty, this is in strong contradiction to the promise of the TJ process (cf. García Pinzón 2020, p. 2; cf. UN). This does not mean that the work or efforts of the Truth Commission are not having an effect, but it is simply not clear whether the truth-finding process has been completed with its final report and what will actually happen with the results now. There is a large discrepancy between the prescribed goal of the CEV and the time provided. In connection with the post-Cold War phase described by Teitel (2003, p. 77 f.), the Truth Commission should aim at reconciliation and thus also a new political beginning (see Section 2.1; Anders and Zenker 2014, p. 395; Wilson 2003, p. 369). Whether a mere listing of serious war and human rights violations is sufficient to fulfill this task is highly doubtful because it is not made clear what a new political beginning will actually look like in order to resolve the existing conflicts peacefully in the future.

### 2.2.2. Special Tribunal JEP

The Colombian *Jurisdicción Especial para la Paz,* or JEP, is the only judicial body within the SIVJRNR. The JEP follows a restorative approach aimed at punishing all those who participated in the armed conflict, either indirectly or directly. The overarching goals of the Special Tribunal are as follows: to provide legal justice for the victims, to present the true events to Colombian society, to protect the right of the victims, to support peace, and to offer legal security to all those who participated in the armed conflict. JEP's mandate is limited to ten years, until March 2028, after which it can be renewed for five more years. If there is further need after the extension, the Special Court may decide to extend it for another five years, so that the maximum possible duration of the mandate is of twenty years (JEP) (See footnote 4).

The JEP assumes two different procedures for seeking justice. The first procedure involves the voluntary acknowledgement of responsibility. Here, participants acknowledge the truth of a matter and accept responsibility for it, and also declare their willingness to make reparations to the victims. In return, no deprivation of liberty is ordered against the acknowledging actors. The second procedure is a legal dispute: that is, a contradictory procedure. After investigation by the *Unidad de Investigación y Acusación*, defendants are charged and tried in a legal process (JEP. Misión, visión, funciones y deberes) (See footnote 4).

Within the TJ process, the special court JEP is the body exemplifying justice. In this regard, the question arises as to whether a judicial determination of truth is sufficient to establish justice. JEP is an important tool, and in terms of a functioning rule of law, it is important for human rights violations to be prosecuted and adjudicated. During my fieldwork in Colombia in 2019 on the topic "The Sexual Reappraisal during the Internal Armed Conflict in Colombia," I conducted semi-structured interviews with different individuals, organizations, and collaborators within the TJ process, such as Ángela Salazar from the Truth Commission. According to Salazar, persons affected by sexual violence[5] during the internal armed conflict have different ideas about how *justice* can be restored. For example,

---

5     In the 1998 Rome Statute, sexual violence as a strategic tool in a war is taxed as a crime against humanity under Art. 7 para. 1 lit. g and as a war crime under Art. 8 para. 2 lit. b. No. xxii as a war crime. According to Art. 7 para. 1 lit. g and Art. 8 para. 2 lit. b. item xxii, sexual violence includes: "rape, sexual slavery, coercion into prostitution, forced pregnancy, forced sterilization, or any other form of sexual violence of comparable gravity."

affected individuals have confided in her that they do not care if the perpetrator in question is convicted. They said it was more important to them that their bodies and genitals be restored. Others want the perpetrators to be convicted at all costs. For almost all of them, it is important that there is no repetition of sexual violence as a weapon of war.[6] Accordingly, linking the results of the different institutions of the TJ process in terms of finding *the truth* and establishing *justice* is important, but should also flow into a public discourse, because without context and public discourse, the results remain invisible.

2.2.3. Special Search Unit for Missing Persons UBPD

The *Unidad de Búsqueda de Personas dadas por Desaparecidas* (UBPD) is a special unit responsible for searching for missing persons who have disappeared in connection with the armed conflict. The mandate of the UBPD lasts twenty years and can be renewed again. The mission of the UBPD Special Unit is to guarantee the return of living persons and, in the case of deceased persons, to determine their identity and deliver their remains to their relatives. The UBPD is not a legal body. The search and identification of missing persons entrusted to it is governed by a national and regional plan (cf. UBPD) ((See footnote 4). According to the UBPD, more than 99,000 people have been reported missing since the internal armed conflict (UBPD, así buscamos).

The Bellavista-Bojayá massacre of 2 May 2002, in the department of Chocó, is an example of the UBPD's special task force. The attack was perpetrated by the former guerrilla group FARC. A cylindrical shell killed 119 people in a church. Earlier, 300 people had sought refuge in this church because the FARC attempted to wrest control of the Rio Atrato basin from the former paramilitary group *Autodefensas Unidas de Colombia* (AUC). No efforts were made to identify the dead or provide medical assistance to survivors until the signing of the peace treaty. With the massacre and the struggle for control of the Rio Atrato area, an estimated 1744 families were displaced and considered internally displaced (Vergara-Figueroa 2018, pp. xix f., 3, 50).

With the identification of the dead and the handover to the family, a burial is made possible. The Bellavista-Bojayá massacre is thus an exemplary example of what the UBPD special unit is used for.

The presented TJ mechanisms of the TJ process in Colombia, i.e., the Truth Commission CEV, the Special Court JEP and the Special Unit for Missing Persons UBPD are used for the determination of *truth* and the establishment of *justice*, so that stable peace becomes viable in a democratic constitutional state. Below, the extent to which the requirements of Roland Dworkin's theory of justice are applicable to a TJ process will be examined. In this context, the concepts of *liberty, equality, democracy, law,* and *truth* and how they are related to *justice*, and how they are to be understood at all in the context of a liberal constitutional state, is of central importance.

## 3. Justice in a Liberal Constitutional State According to Dworkin

Roland Dworkin's theory of justice refers to a successful conduct of life, as it was already of significance in antiquity in Aristotle's works and, in terms of historical influence, also with theologians of the Middle Ages, such as Thomas Aquinas. Accordingly, a successful conduct of life is not possible without interweaving values and without *truth* in relation to facts of life. Rather, an ethical attitude is necessary (2014, pp. 13, 207). According to Dworkin, the two ethical principles of *self-respect* and *authenticity* are elementary in order to achieve a successful way of life (2014, p. 346). *Self-respect* means respect for one's own person and the recognition of an independent concept of life that leads to a successful conduct of life (Dworkin 2014, p. 348 f.). *Authenticity* is understood in terms of how we shape life and what basic conscious attitude is adopted (Dworkin 2014, p. 356 f.). *Self-respect* and *authenticity* are to be considered together with human dignity and the equality of people, which are indispensable within a liberal constitutional state according

---

[6] The statement comes from the interview with Ángela Salazar in Bogotá on 12 April 2019.

to Dworkin (2014, p. 345 ff.). However, the theory of justice does not work without the interpretation of values for the individual person as well as for society. This is due to the concept of justice in the today's dynamic society being constantly subject to social and political changes and these developments having an influence on the life of each individual person as well as on society as a whole (Dworkin 2014, pp. 269, 307).

One can refer to the verse of the ancient Greek poet Archilochos about the fox and the hedgehog, which is the eponym for Dworkin's theory of justice.[7] The quote in question is: "The fox knows many things, but the hedgehogs knows one big thing. [*Truth*] is one big thing" (Dworkin 2011, p. 1, Quoted from: Isaiah Berlin (2009), The Hedgehog and the Fox. Essay on Tolstoy's Understanding of History, Frankfurt/M.: Suhrkamp 2009, p. 7). According to Dworkin, *truth* thereby cumulates through an interconnected network of values that together form a supporting togetherness. Dworkin argues that *truth* should be seen as an "interpretive assertion" so that *truth* can be discussed (2014, p. 296).

According to Dworkin, the scientific interpretation is to be understood in the sense of an *active holism,* meaning the dense interweaving of values as a whole. Thus, the interweaving of individual values with others is the underlying idea. Consequently, a change or questioning of one value always affects the whole network of values. Thus, an interpretation is always active, which is Dworkin's (2014, p. 263 ff.) consequent demand from scientific interpretation. Furthermore, values are to be considered as equal and accordingly have no hierarchical function among each other. Accordingly, the terms *liberty*, *equality*, *democracy*, *right* and *truth* are also equivalent in the context of *justice*. Consequently, justice is to be understood as an independent value, which is however connected with different values and thus also affected by these. According to this assertion, Dworkin's theory of justice always depends on the interpretive approach of how values are interpreted. Accordingly, Dworkin's theory of justice is to be understood as a dynamic one that is influenced by the social contexts examined.

If Dworkin's theory of justice is applied to the TJ process, specifically to the TJ process in Colombia presented earlier (see Section 2.2), the following considerations can be made. The TJ process in Colombia has as its overarching goal to establish stable *peace*, for which various mechanisms and bodies have been created (see Section 2.2; cf. JEP). According to Dworkin's theory of justice, one could deduce that *peace* can only exist in a democratic state based on the rule of law, which is characterized, among other things, by the interwoven values of *liberty*, *equality,* and *justice*. In this context, however, it must be remembered that democratic states also wage or support war in order to protect *democratic values*. Accordingly, *liberty* is not to be understood as a fixed constant. Values, as well as a TJ process, are at the mercy of a dynamic process, and the establishment and preservation of *peace* is to be understood as a negotiation in constant flux (see Section 2.1). Accordingly, the stability of *peace* cannot be understood as static because peace itself is subject to constant change, subject to social, political, and economic conditions. In addition, there is also a discordant attitude toward peace policy in Colombia, which does not automatically presuppose unity for peace policy due to conflict-ridden disputes. Therein lies the greatest challenge in a TJ process: finding solution that takes into account *all* voices and thus reflects the values that are shared by *all*. In this regard, it is critical to consider whether this ideal is at all feasible.

*3.1. The Interpretive Approach*

Values are to be fathomed as being part of a multilayered and intertwined network connected to other values. The interpretive approach should therefore be understood in relation to *active holism*. It is considered to be an interpretive reasoning in relation to the interlocking of values. Thus, an interpretive assertion is never simply true, but addresses a particular event to provide the best possible interpretive justification. Value concepts change and new knowledge alters the big picture, so interpretive statements must be

---

[7]　Dworkin quotes in his work from the book by Isaiah Berlin, Der Igel und der Fuchs. Essay on (Tolstoy's Understanding of History, Frankfurt/M.: Suhrkamp 2009, p. 7; Dworkin 2014, pp. 13, 715).

constantly renegotiated (Dworkin 2014, p. 263 f.). Accordingly, *justice, liberty, equality,
democracy, law*, and *truth* are also interpretive concepts that are subject to permanent change
and are deemed pillars of a liberal constitutional state. The right to vote can be mentioned
as an example in this context. With the formation of nation states in the 19th century, the
right to vote was attributed mainly to men. The demand for *equality* in a liberal state thus
related primarily to the participation of men (cf. Senn 2020, p. 7). In this respect, the
demand for *equality* and *justice* in a liberal state under the rule of law is not to be simply
taken for granted, but requires a continuous questioning of what we understand by these
values. With regards to interpretation, Dworkin (2014, p. 172 f.) distinguishes between
*moral responsibility* and general as well as *conceptual interpretation*. *Moral responsibility* refers
to an integrated moral epistemology that relates to sound reasoning about moral issues. In
this regard, *moral responsibility* is related to the question of integrity (Dworkin 2014, p. 174).
The issue of integrity, as well as *moral responsibility*, refers to *individual responsibility* on
the one hand, but also to how a community makes political decisions on the other. The
problem of resource distribution or equal treatment of citizens can be subsumed under this
(Dworkin 2014, p. 601 f.).

In Dworkin's (2014, p. 214) *general interpretation*, the theory of value is central to
address general conditions of interpretation. Thereby, according to Dworkin, there is no
generally valid interpretation that can be applied in all fields, but there has to be a case-by-
case assessment from the presuppositions of the different disciplines. An interpretation is
always influenced by the usual standards, such as a scientific methodology or by concrete
social practices in everyday life (Dworkin 2014, p. 22 ff.). Accordingly, the justification of
interpretations is only possible through an extended interpretation. Thereby the person
who interprets bears a responsibility, i.e., it is central how a concrete situation is suitably
represented by means of an interpretation (Dworkin 2014, p. 22 ff.). According to Dworkin
(2014, pp. 205, 225), concepts are to be interpreted again and again, because the shared social
practices change and are to be understood within the respective zeitgeist. This includes, for
example, a moral value conception, which should correspond to the basic ethical attitude
of the interpreting person(s) (cf. Derrida 1972, p. 423; cf. Senn 2017, pp. 6 f., 9; cf. Senn
1993, p. 73 f.).

*Transition* and *justice*, according to Jens Ohlin (2007, p. 51, see chap. 2.1) are also
interpretive terms in the Dworkin sense, because they are moral and political concepts. TJ
revolves around universal values such as human rights and about a respective political
state of exception characterized by violent conflict. Thus, the universal values of human
rights cannot be considered in isolation and implemented in a society without contextu-
alizing the political and conflictual context. Consequently, any interpretation requires a
contextualization of the problem linked to it, which, according to Dworkin (2014, p. 24),
is based on interpretation, which in turn consists of values. It is for this reason that a
permanent discussion, as it is maintained by philosophers, is paramount.

Accordingly, the fundamental problem of a TJ process in general can be defined as
follows. On the one hand, TJ relies on the universal value of *justice*. On the other, it
is confronted with social and context-specific events that call into question a universal
solution approach. A universal solution approach is hardly feasible if one wants to do
justice to the different conceptions of *justice*. Although the problem appears to be universal,
it still cannot be solved universally, but rather concretely.

In the context of the TJ process in Colombia, it will now be shown how an integrated
moral epistemology feeds into the process. The question to ask is how, in a TJ process, the
conditions for *individual responsibility* (with regard to a successful way of life) and *collective
responsibility* (with regard to political decisions) can be realized. It is essential that the
concepts of *justice, liberty, equality, democracy*, and *right* are understood as interpretive in the
TJ process because they are linked to concrete social and political conditions.

In what follows, I shall use the interpretive approach and selected perspectives from
the philosophy of law, social philosophy, and social anthropology to show how interpreta-
tions and attributions of meaning are dealt with in different disciplines.

### 3.1.1. Scientific Self-Reflection and Sovereign Creative Capacity

Questions about *justice* and humanity in a social and cultural context are a central subject of the philosophy of law and society. In connection with the historical-critical examination of different understandings of law and their historical influence, it provides a suitable basis for considering the concept of TJ in the framework of Roland Dworkin's theory of interpretation and justice (cf. Gyr 2020, p. 156).

*Open and clear conditions* are necessary to maintain a critical debate in the discipline and thus to enable a scientific self-maturing process. *Scientific self-reflection* is understood as requiring a critical examination of one's own collective discipline as well as one's own *individual* ways of thinking and attitudes, insofar as these are always shaped by a social and cultural worldview that feeds into scientific text production (Senn 1993, p. 72 f.; vgl. Gyr 2020, p. 162 f.). Dworkin's (2014, pp. 22 f., 679) interpretive approach is about recognizing the social, economic, and political practices that influence our thinking and discipline and that feed into interpretations.

Historiography is read and interpreted depending on the zeitgeist and the way of thinking and can thus never be reduced to the same denominator (Stolleis 2016, column 1497; cf. Senn 1993, p. 73). Thus, it takes a kind of *sovereign creativity* to present the results of a historical analysis. The writing of scientific texts is thus always a construction that can never represent the whole picture (cf. Gadamer 1975, p. 465). Within *sovereign creativity*, the goal is not only to produce a constant improvement of knowledge, but also to show new points of view. It is about reflexive and conscious interpretation of history based on a critically examined experience and how it can be transposed into the present (Senn 1993, p. 73 f.).

In relation to the TJ process in Colombia, which is currently ongoing, the question is what results contribute to how the story about the internal armed conflict in Colombia is presented for the future. The TJ process in Colombia—SIVJRNR—is tasked with reporting on the true facts of the conflict in order to prevent recurrence of the conflict and its grave human rights violations (see Section 2.2). The Truth Commission CEV has a special role in terms of truth-telling. However, the time allotted for truth investigation is too short. In this context, it is critical to consider how it is possible to adequately resolve a conflict that has lasted more than fifty years within three years (see Section 2.2.1, cf. CEV, Mandato y Funciones) (See footnote 4). In addition to the CEV, the Special Court JEP and the Special Unit for the Search for Missing Persons UBPD are also used to determine the truth (see Section 2.2). The JEP refers to legal truth-finding and the UBPD to the search for missing persons. The latter is primarily concerned with whether the missing persons are still alive, whether they disappeared in the context of the internal armed conflict, or whether the persons in question themselves are to be counted as members of parties to the conflict or as civilian victims (see Section 2.2.3; cf. UBPD) (See footnote 4).

The TJ process in Colombia highlights the tension between *transition* and *justice*. On the one hand, there are institutions such as the CEV, JEP, and UBPD that have been installed to determine *the truth* to achieve *justice*. On the other hand, they are bound by political circumstances as to exactly how *truth* can be determined. For example, there is a lack of *open and clear conditions* on how affected persons can participate in the process at all. Non-governmental organizations (NGOs) play an important role in mediating between affected persons and the institutes of the TJ process (see Section 2.1). At the time of my field research in 2019, the clear conditions governing how exactly individuals or a collective can participate in a TJ process had not been fully determined. In addition, a list of individuals who provided testimony was published during this time by mistake, which did not promote confidence in the TJ process (See footnote 4). In addition, in relation to historical contexts, a rejection of the parties to the conflict continuing with serious human rights violations is notable. For example, no party to the conflict wants to be associated with the use of sexual violence as a weapon of war. In this regard, an attempt is being made to focus on *individual responsibility* as a result of having been a party to the conflict that used sexual violence as a tactical weapon of war hampers the process of electioneering in search of a new beginning.

3.1.2. The Perspective of Social Anthropology

Social anthropology[8] emerged during the colonial period of the 19th and early 20th century and aimed at systematically researching "foreign peoples" and describing their cultural habitat as comprehensively as possible (Kohl 2012, p. 131; cf. Gyr 2020, p. 163). In this context, so-called "foreign cultures" were regarded as objects of research that could be objectified. During this period, the theory of evolutionism also played an important role; it was intended to trace the progress of mankind in the field of society and culture and to legitimize the colonial masters vis-à-vis the subjugated and, in the view of the colonial masters, "second- or third-rate" peoples (Kohl 2012, p. 154; cf. Gyr 2020, p. 163 f.; cf. Senn 2017 with regard to "othering", p. 199; cf. Senn and Gschwend 2010 with regard to "racial doctrine", p. 86). Racial doctrine rapidly gained popularity in academia as the answer to the inequality between the "superior" and "inferior races" lay in the responsibility of the bearer of a race itself (Kohl 2012, p. 154 ff.). Racial doctrine must also be considered along with the politics and nation-building of the 19th century. The political focus was on finding a national identity based on a common denominator, namely one's own people. In this regard, the problem and danger of a scientific discussion of cultural differences should also be noted, especially when it can be misused for political purposes (Senn and Gschwend 2010, pp. 77 f.). Racial doctrine also aimed at justifying colonial policy, and the task of cultural anthropologists was to show the differences between cultures and thus legitimize the power of the "superior race" (Senn and Gschwend 2010, p. 86; cf. Senn 2017 about "othering", p. 199). For this reason, anthropology in Europe was tied to the system of colonialism, and with the collapse of this structure, interest in the discipline also declined, making it necessary for anthropologists to come to terms with their own discipline.

The most impactful debate within social anthropology took place in the late 1970s and is known as the *Writing Culture Debate* (Marcus 2012, p. 73; cf. Gyr 2020, p. 164). The debate addressed the colonial legacy and the question of how to write about cultures. Ultimately, it was also about demonstrating the presence of colonial structures within social sciences, especially in social anthropology and more specifically in the context of researching the "foreign" (Beer 2008, p. 13; cf. Gyr 2020, p. 164). The critique focused primarily on ethnography, which emerged out of the confluence of field notes, observations, dialogues, and discourses in written form (Marcus 2010, p. 264 f; cf. Gyr 2020, p. 164). The central problem of ethnography was (and still is) that from an interpretation—without further reflection—a neutral objective attribution of meaning was assumed and thus the interpretive result had to be regarded as generally scientifically valid. This problem can therefore be compared to the problem of positivism (cf. Gyr 2020, p. 164).

According to Jacques Derrida (1972), social anthropology occupies a privileged sphere of action solely because that the discipline lost its privileged position with the collapse of colonialism, and it then had to deal with its own raison d'être. By trying to break away from Eurocentrism, it simultaneously absorbed the idea of Eurocentrism. According to Derrida (1972, p. 427), this cannot be circumvented, because only by acknowledging historical contexts can they also be consciously and seriously questioned (cf. Dworkin 2014, p. 205: cf. Senn 1993, p. 73 f.).

The temporary structures which TJ as well as positivism can be examined by means of the "structurality of structure" as developed by Derrida. According to Derrida (1972, p. 422), the "structurality of structure" in science serves to reduce a structure until a core emerges that reveals a steady origin from which all attributions of meaning can in turn be derived. According to Derrida (1972, p. 422), however, the center is located both inside and outside the structure. Thus, events are characterized by realizations of a respective zeitgeist, whose beginning and end must be renegotiated time and again. This means that the attribution of meaning has a complexity that cannot simply be limited to a structure

---

8    In the following, no distinction is made between the application of terms of social anthropology, cultural anthropology, social and cultural anthropology, and ethnology. All areas deal with the same problem, but with different approaches and accentuation, whereby the differentiation is not addressed in this study.

and a center. The same is true of Dworkin's (2014, pp. 269, 307) interpretive theory with respect to the interpretation of value ascription. In this sense, positivism in social science as in jurisprudence is always only one way to find *truth*. For positivism severely limits perspectives from the outset by its self-imposed framework, so that the reconstructed truth taken as a basis here is always only able to reflect a small and probably also distorted section of history (cf. Gyr 2020, p. 159 f.).

In the context of TJ, the events in question are mostly violent conflicts. Nevertheless, through the temporary structure of TJ, *the truth* is to be determined and *justice* is to be established. However, with neutralization and reduction, socio-cultural differences are filtered out of the process. This allows for a universalism that does not exist in the practice of TJ in this way (cf. Gyr 2020, p. 159 f.).

With respect to the TJ process in Colombia, the issue of politics and nation-building in the context of racial doctrine is quite relevant from a social anthropological perspective (see Section 3.1). Most Latin American states gained their independence from colonial states in the 19th century. As a result, especially in the second half of the 19th century, laws were enacted in the new states that allowed an elite to appropriate land from indigenous communities and privatize communal property (Huizer and Stavenhagen 1974, p. 379 f.). According to Anibal Quijano (2000, p. 533 f.), for example, racial doctrine was deliberately used as an instrument of power during the period of decolonization (cf. Senn and Gschwend 2010, p. 86). Alyson Brysk (2000, p. 7); they argue that the distribution of land from collective to individual ownership was detrimental to societies in Latin America because it gave rise to elites with whom a majority of the population could not identify. According to Quijano (2000, p. 534 f.), the identities created by the colonialists, such as *Indios* and *Afros*, have erased previous identities and histories, reducing collective identities to the event of colonization. Thus, the people referred to as *Indios* by the colonialists are those who were present before the colonialists and the *Afros* are the slaves who were sold to the Americas to increase global labor production.

With regard to the TJ process in Colombia, it can be stated that mainly regions where *indigenous* and *Afro-Colombian* people live were, or still are, affected by violent conflicts. During my 2019 fieldwork in Colombia, I traveled to Bogotá and the department of Chocó. Bogotá is home to all the headquarters of the institutionalized mandates of the TJ process, such as the Special Court JEP, and the department of Chocó is one of the departments in Colombia that is still contested by different parties to the conflict. The discrepancy between Bogotá and Chocó is demonstrated not only by the violent confrontation but also by the absence of rule of law. Consequently, the TJ process in Chocó is not represented by an institution. It is mediated primarily through NGOs, which are networked locally and internationally and are thus gaining in importance (Teitel 2003, see chap. 2.1, *Post-Cold War Phase*).

### 3.2. Liberty and the Rule of Law as the Basis of Transitional Justice

To achieve *justice* in a constitutional state, a theory of *liberty* is required so that the framework between the government and the people is clarified. Dworkin distinguishes between *Freedom* and *Liberty*. By *Freedom*, he means the *freedom* to do anything and everything without any restriction from the government. *Liberty*, on the other hand, entails certain rights of *freedom*. In the context of a liberal constitutional state, the discussion always pivots around certain *liberties* and not about the *freedom* to do everything without government restriction. However, there is a great deal of leeway in this regard, which is constantly being changed between the government and the population. In this context, *liberty* often conflicts with the question of *equality* (Dworkin 2014, p. 18). This conflict is also observable in Colombia's TJ process, such as in relation to natural resource allocation.

### 3.2.1. Liberty According to Dworkin

The general idea of *liberty* is widespread and known, but what is of interest in this context is the connotation of *liberty* in the context of a just state. According to Dworkin (2014, p. 616), to classify the concept of *freedom*, it must be treated interpretively so that the

controversy can be addressed at all. In a liberal constitutional state, *liberty* is associated with coercion, which, according to Dworkin (2014, p. 617), must be considered in the context of personal responsibility. Personal responsibility here consists of the two basic ethical principles of *self-respect* and *authenticity*. This entails that it is the responsibility of every human being and, accordingly, their *liberty* to appreciate and decide by making "something" of their own lives (Dworkin 2014, p. 345 f.). Taking responsibility for one's life, then, means living life in such a way that it conforms to self-imposed ethical values (Dworkin 2014, pp. 346, 357). Transferred to the state community, political decisions are subject to conditions related to respect for *individual responsibility* (Dworkin 2014, p. 617). This means that participation in collective decisions must be guaranteed and that decisions concerning personal responsibility can only be made by the actual people affected by them (Dworkin 2014, p. 618).

Regarding *liberty*, Dworkin (2014, p. 618) distinguishes between *positive* and *negative liberty*. *Positive liberty* is concerned with what appropriate means can be used to ensure the necessary participation of citizens. *Negative liberty*, on the other hand, encompasses the situations when collective decisions come into conflict with personal responsibility.

According to Dworkin (2014, p. 621), controversies about *liberty* can only be thoroughly grasped if we understand the concept of liberty interpretively and relate it to personal responsibility. Only in connection with human dignity does *liberty* appears valuable.

Accordingly, all people should be entitled to the same *liberty,* which is why the implementation of equality is crucial. In the constitutional state, the question arises under which conditions a government can justifiably restrict the *liberty* of all and where the limits for this lie (Dworkin 2014, p. 624). Ethics plays an important role when it comes to drawing the line between collective decisions and *personal responsibility* (Dworkin 2014, p. 628). According to Dworkin (2014, p. 33), ethics is related to the topic of leading a successful way of life, whereas morality focuses on how we behave towards other people. Accordingly, for Dworkin, a successful way of life is elementary in terms of *liberty* and responsibility. The individual thus plays an important role according to Dworkin. Regarding responsibility, the focus in a TJ process is on the conflict parties as a collective. In this context, *collective responsibility* in a TJ process refers to acts of serious human rights violations committed during the internal armed conflict. Personal responsibility, as described by Dworkin, is to be understood in the framework of a functioning liberal state under the rule of law that is not at war or in armed conflict. The description bears a discrepancy between *personal* and *collective responsibility*. According to Dworkin, *personal responsibility* should adhere to basic ethical principles. In relation to a violent conflict, goals, which can also be related to the idea of a successful way of life, are enforced by violent means. Therefore, it is not clarified what happens when the framework conditions that enable the fundamental values of a successful way of life are missing and how these are built up, which in turn is precisely the goal of a TJ process.

### 3.2.2. Transitional Justice and Liberty

*Liberty* in the aforementioned sense is also a central element in a TJ process because the result is to guarantee *Liberty* (see Section 2.1). Regarding a TJ process, the question arises whether personal responsibility in the sense of *self-respect* and *authenticity* is guaranteed.

In a TJ process, the relationship between citizen and state is complicated, as shown by the example of the peace treaty in Colombia (see Section 2.2.1). It is therefore difficult to comply with the conditions described by Dworkin regarding *liberty*. The two basic ethical principles of *self-respect* and *authenticity* are in a state of exception and subject to difficult, unpredictable conditions. Particularly, it is unclear to what extent a person in a state of emergency of violent conflict can participate in the decisions of a collective and to what extent respect for personal responsibility can or may be taken into account in these decisions (cf. Dworkin 2014, p. 618). In the TJ process of Colombia—SIVJRNR—there is no indication of how the idea of liberty of individuals can be concretely implemented. The TJ process aims to shed light on past events; this is done through a report by the

Truth Commission, the search for missing persons and the Special Court (see Section 2.2). However, the SIVJRNR does not provide a concrete indication of how the relationship between the citizen and the state will be shaped in terms of the two fundamental ethical principles. The TJ process provides mechanisms that are used primarily for dealing with the violent events of the past. There are no specific instructions on how the *liberty* and *equality* of Colombians should be shaped in concrete terms. For years, Colombia has been one of the countries with the most internally displaced persons. According to the United Nations High Commissioner for Refugees (UNHCR), between June 2021 and May 2022, approximately 60,000 people were internally displaced or forcibly resettled (UNHCR 2023). People in rural areas, such as the departments of Chocó, Cauca, Nariño, and Norte de Santander, are particularly affected. Although the report of the Truth Commission points out this situation, it cannot take any measures other than making recommendations (see Section 2.2.1). Here, the tension between "transition" and "justice" is clearly manifested (see Section 2.1). The discrepancy between an ideal conception of liberty and the actual political circumstances is substantial. Liberty, in terms of personal responsibility and political co-determination, depends on conditions that make it possible to participate in a process in the first place and to be respected as equal participants. The decisive factor here is the value attributed to freedom and how it connects with equality. If no values were ascribed to liberty in a political decision-making process, it would be meaningless and the people would likewise have no significance.

### 3.3. Equality

In addition to *liberty*, *equality* is another factor in achieving *justice* within a constitutional state. As already mentioned, the question of equality can come into conflict with the idea of *liberty*. According to this, the goal is the equal treatment of people within a constitutional state, so that everyone decides for themselves how to shape their lives (Dworkin 2014, p. 15). However, the self-determined shaping of one's own life depends on the possibilities and how resources are distributed within a state. This means, for example, access to education and state infrastructure, but also the distribution of and access to land. The challenge in the distribution of resources is to ensure that people are treated equally (Dworkin 2014, p. 15 ff.). Accordingly, the question is how decisions of a political community can bring the distribution of resources in line with personal responsibility: namely, how the individual life design with collective, political decisions can meet the requirement of justice for liberty and equality (Dworkin 2014, p. 602). In relation to this, it is not clear how a state distributes natural resources by appropriate means that meet the demand for *equality* and *liberty* at the individual and collective levels, which are also in a constant state of flux (cf. Dworkin 2014, pp. 269, 307, see chap. 3).

### 3.3.1. Equality According to Dworkin

In resource allocation, Dworkin (2014, p. 601) distinguishes between *personal* and *impersonal* resources. The issue of resource allocation by a political community relates only to impersonal resources. According to Dworkin (2014, pp. 697 f., 600 f.), an equitable distribution of resources is not possible in a *laisser-faire* state, nor in a pure *welfare* state. Dworkin argues they both fail under the principle of *distributive justice*. In a *laisser-faire* environment, Dworkin notes (2014, p. 597 f.), the state does not assume the responsibility of a fair distribution of resources because it behaves passively. He sees (Dworkin 2014, p. 600 f.) the problem with a *welfare* state, on the other hand, in that it determines what constitutes a good life and a successful lifestyle; thus, personal responsibility cannot be guaranteed.

As claimed by Dworkin (2014, pp. 15 ff., 549 f.), the allocation of resources by the government should be transparent and based on the principles of equal treatment and individual responsibility; the state must bear responsibility so that domestic power-political interest groups cannot disproportionately enrich themselves. Dworkin's (2014, p. 601 f.) concept of equality is about the consideration of all citizens within a political community.

On the one hand, the personal responsibility of all members should be respected and, on the other, political decisions about resources should also be made on a basis of ethical responsibility.

3.3.2. Resource Equality and Transitional Justice Process in Colombia

With regard to the internal armed conflict in Colombia, the distribution of resources can be described as a major problem. Overall, the political disputes between liberals and conservatives in the 1940s and the social differences in the population are considered to have triggered the emergence of different guerrillas, which demanded a redistribution of power and land claims (Farnsworth-Alvear et al. 2017, p. 343). The emergence of drug trafficking in the 1980s and 1990s tightened the web of conflict (Richani 1997, p. 37; Werle and Vormann 2018, p. 286). The department of Chocó is one of the main hotbeds of the conflict of interests described above. The Bellavista-Bojayá massacre of 2 May 2002, in the department of Chocó exemplifies the massive use of violence to gain power and control over a catchment area on the Rio Atrato (see Section 2.2.3; Vergara-Figueroa 2018, pp. xix f., 3, 50).

The example of the Rio Atrato can be used to show different constellations, as it is central to the social, political, economic, and historical self-understanding, and to show the different conflicts and consequences for the population in the Chocó. The river was recognized as an independent legal entity by the Colombian Constitutional Court on 10 November 2016 (Río Atrato. Sentencia T-622/16). With this ruling, the river is recognized as an important source of livelihood for the department of Chocó. The river not only stands for biodiversity, but it is also the livelihood for the population in Chocó in general, a trade route, and attraction for gold prospectors (Vergara-Figueroa 2018, p. 27 f.) With the aforementioned ruling, the State of Colombia was obliged to ensure the protection of the river and its associated biodiversity.

According to Vergara-Figueroa (2018, p. 3), the colonial past continues to play an important role as people in the department of Chocó still receive little attention within the government. Access to state infrastructure is almost non-existent in the department, which speaks for an unequal distribution of resources. The Colombian government does not assume any responsibility in this regard. It is critical to consider how voices from Chocó are taken into account in political decisions related to resource distribution. According to Dworkin's (2014, pp. 15 ff., 594 f.) theory, this is necessary in order to live a successful life (see Section 3.3.1). The fact that violence has flared up again in the department of Chocó since the peace treaty shows that a peace treaty does not automatically mean peace.

The problem is that the peace treaty was concluded only between the government and the former guerrilla group FARC (see Section 2.2.1; cf. Werle and Vormann 2018, p. 287). Consequently, the entire drug conflict and the conflict between guerrilla groups, such as the *Ejercito Nacional de Liberación* (ELN), have not been included in Colombia's TJ process. Therefore, it matters in which department of Colombia someone lives. It is so difficult to treat citizens equally as, depending on the department, people live in different conditions and are affected differently by the conflicts that still continue. It is obvious that the Colombian government is far from ensuring equal treatment of the people of Colombia when it comes to the distribution of resources, because the different ideas of a successful way of life when making political decisions are not taken into account, and control cannot be asserted over the entire national territory.

*3.4. Democracy*

A liberal constitutional state is based on a democratic idea of the state. In a liberal constitutional state, Dworkin (2014, pp. 642, 647) understands the idea of *democracy* in the relationship of *equality*, *liberty,* and *justice*. *Liberty* and *equality* can come into conflict with each other when it comes to resource distribution, for example (see Sections 3.2 and 3.3). *Democracy* means is often associated to the form of participation in government and the guarantee of access to state power (Dworkin 2014, p. 19).

In a TJ process, equal participation plays an important role in ensuring that political decisions are supported as broadly as possible by the population. The question is how the concept of *democracy* is negotiated in Colombian society and whether the decision to no longer bring the revised peace treaty before the people satisfies the demand for *justice* (see Section 2.2).

### 3.4.1. Democracy According to Dworkin

In a *democracy*, according to Dworkin (2014, p. 653), human dignity is central to creating conditions in a state that allow people to be treated equally. Political equality is crucial to how political power is distributed. Dworkin (2014, p. 657) is not concerned with a mathematical calculation of how power is distributed; rather, his point is that the actual distribution of power also leads to equal treatment of citizens. *Democracy* is often described as a form of government by the people. Since the concept of *democracy* is interpretive, it is unclear what is meant by *the people*. Furthermore, it is not completely clear how political participation is guaranteed, since elections depend on the respective electoral systems, which in turn can have different influences on the final result (Dworkin 2014, p. 641 f.). *Democracy* therefore presupposes a political community, but says nothing about how this political community should be composed (Dworkin 2014, p. 643). It is therefore crucial what conditions are created for a person to belong to a political community. In relation to this, it is important to look critically at what happens to people who are not counted as part of a political community, but who are nevertheless affected by collective decisions. This includes, for example, stateless persons or sans papiers.

In a *democracy*, collective decisions can also be enforced by coercion. According to Dworkin (2014, p. 617), any government is coercion-based when it comes to negotiating the framework conditions. This is where the theory of *liberty* comes in, according to which Dworkin (2014, p. 18) is primarily concerned with certain rights of *liberty* (see Section 3.2). Dworkin (2014, p. 641) refers to a fundamentally ethical attitude as decisive, which goes hand in hand with governmental and individual responsibility. This means that every person is called upon to make ethical decisions independently of the government.

### 3.4.2. Transitional Justice in Colombia: Democracy and Belonging

The TJ process in Colombia was initiated by the signing of the Colombian government and the former guerrilla group FARC in November 2016, which at the same time reflects the ambivalence of the population regarding the peace policy (see Section 2.2; Werle and Vormann 2018, p. 287). A TJ process is about creating democratic state structures to ensure equal treatment of people and political participation (see Section 3.4.1). In this regard, it is not clear who is counted as part of the political community because *democracy* is an interpretive term (see Section 3.4.1; Dworkin 2014, p. 641 ff.). In this context, it is important to consider how the TJ process in Colombia interprets the concept of *democracy* in the first place and who is counted as part of the political community.

The Truth Commission takes up the question of who belongs to the people because one task concerns "national reconciliation" (Wilson 2003, p. 371). "National reconciliation" must be viewed critically in relation to the founding of the state of Colombia and the concomitant nationality and identity formation in the 19th century and the re-nationalization and identity politics of the TJ in the present (cf. Gyr 2020, p. 168 f.; cf. Senn and Gschwend 2010, p. 77 f.). The 19th-century search for nationality and identity can be compared to the emergence of social anthropology in the colonial context. In this setting, evolutionary theory and racial doctrine were important tools to legitimize domination over "third- or second-rate peoples" (see Section 3.1.2; cf. Kohl 2012, p. 131; cf. Gyr 2020, p. 163; cf. Senn 2017, regarding "othering", p. 199). It was only through a serious and prolonged engagement with the colonial legacy, triggered by the *Writing Culture Debate* in the late 1970s, that social anthropology could emerge anew (see Section 3.1.2; Derrida 1972, p. 427; cf. Gyr 2020, p. 164; cf. Dworkin 2014, p. 205; cf. Marcus 2012, p. 73; cf. Senn 1993, p. 73 f.).

In the context of the TJ process in Colombia, it is first necessary to examine whether nationality and identity politics play any role at all in achieving the goals. According to Huizer and Stavenhagen (1974, p. 379 f.), with the independence of the colonial states in the 19th century, an elite was formed which, through new laws, made it possible to appropriate land from indigenous communities and to privatize communal property (see Section 3.1.2; Brysk 2000, p. 7). In doing so, according to Quijano (2000, p. 533 ff.), racial doctrine was deliberately used as an instrument of power, and identities such as *Indios* and *Afros* were created by the colonialists, who reduced their identity to the event of colonization (See Section 3.2.1). At an event of the TJ process, which provided information about the participation of affected persons and groups of the internal armed conflict in Colombia, there were also meetings of different communities of interest. In relation to reconciliation, for example, the testimony of a person who was a representative of a religious group is that the reconciliation should also end the territorial claims of the indigenous population. He added that the dignity of the victims should be central and that a public apology was needed, but that indigenous people are not entitled to land. He feared that the latter were aiming to present themselves as victims. However, in the context of this event and the different representatives of communities of interest, it clearly emerged that opinions differ significantly on the idea of what a "reconciliation" should involve (See footnote 4).

In Colombia, slaves were used for forced labor and land was privatized from indigenous people. Around the middle of the 19th century, slavery was abolished in Colombia. The first law that allowed Afro-Colombian communities to own land was the so-called Law 70 of 1993 (Ley 70 de 1993; Vergara-Figueroa 2018, Foreword). Law 70 concerns Afro-Colombian communities and protects their ethnic identity and the preservation of traditional ways of life, with the aim of providing them with equal opportunities (Artículo 1, Ley 70 de 1993).

With this law, it became possible for these communities to make a collective claim to land. At the same time, the law is linked to a notion of identity, which it thereby seeks to protect and which can thus also only be asserted collectively. It is questionable how traditional ways of life are determined and what happens to communities that do not meet the requirements. In addition, there is a discrepancy that arises from the pressure to conform to Colombian society and the right to land, which is only granted to those who lead a "traditional" way of life. According to Jelin (1996, p. 105), this can be understood as a marginalization of a group that attaches rights to identity that are only guaranteed collectively.

In reference to Dworkin's (2014, pp. 263 f., 269, 307) interpretive approach, values are subject to constant change and require permanent negotiation (see Sections 3 and 3.1). Consequently, even a "traditional way of life" cannot be regarded as a static and unchanging way of life that does not change. Accordingly, collective rights that demand a "traditional way of life" are not compatible with personal responsibility, according to Dworkin (2014, p. 346, see chap. 3). The requirement of a "traditional way of life" in order to assert certain rights contradicts this principle. At the same time, it is also unclear how a "traditional way of life" can be guaranteed in terms of equal treatment and participation in political decision-making (cf. Dworkin 2014, p. 653). Due to the ongoing conflicts as well as the apparent ambivalence of the population regarding the peace treaty, Dworkin's demands have not been met.

*3.5. Law*

*Law* is the decisive factor for the rule of law. *Law* is not simply an instrument of a government; in fact, the government itself is subject to law. Only autonomous law as the basis of a state can permanently guarantee a peaceful and secure coexistence of a society, if the autonomy of *law* is also guaranteed (Senn 2022, p. 3). The examination of a society's understanding of its *law* is also an examination of its history. Within a TJ process, for example, it is necessary to deal responsibly with the history to be processed in order to get to *the truth* of what happened during the armed conflict (cf. Senn 2022, p. 2).

According to Dworkin, *law,* along with *equality, liberty,* and *democracy,* complements the demand for *justice.* Dworkin understands *law* as part of morality, which he considers a "tree-like structure":

> "It is also necessary to understand morality in general as having a tree structure: law is a branch of political morality, which is itself a branch of a more general personal morality, which is in turn a branch of a yet more general theory of what it is to live well." (Dworkin 2011, p. 5.)

It has not been fully clarified what role *law* should or can play within TJ. In legal theory, *law* assumes an integrating function for a functioning constitutional state, such as for peacekeeping and social integration (cf. Habermas 1992, with regard to "social integration", p. 527; cf. Mahlmann 2018, with regard to "functions of law", pp. 32–37).

Concerning TJ, however, it is not the state that forms the framework structure of law; rather, the fundamental orientation of law is located at the international level, as was already the case, for example, in the guiding ideas of natural law doctrine from antiquity through the Middle Ages and into modern times (cf. Welze 1962). Consequently, it is also unclear to what extent societal contexts and basic ethical attitudes are incorporated into the process of the different mechanisms of TJ in order to obtain an integrating legal function.

### 3.5.1. Legal Understandings

The philosophy of law and society, as formulated by Marcel Senn (2017, p. 4 f.), involves a historical-critical examination of the understanding of law per se and in its respective social reception, and sheds light on how it affects or affected a society (cf. Gyr 2020, p. 160). The plurality of views regarding *law* within a society is also addressed. For this reason, we also speak of philosophy of law and social philosophy, and not only of philosophy of law (Senn 2017, Foreword, p. V; cf. Gyr 2020, p. 160).

*Law* is to be understood within the respective social and cultural environment of a society and can be compared with Dworkin's interpretive approach.

A legal and socio-philosophical perspective deals with a historical-reflexive discussion and with how different conceptions of law can be examined.

### 3.5.2. Transitional Justice and Principles of General Natural and International Law

Concerning Dworkin's theory of interpretation and justice, the shaping of law and the idea of a liberal constitutional state must be understood in the context of the North American and European reception history. With regard to TJ, it is not a state that forms the framework structure of law, but TJ takes place in an international context and can be compared with the principles of general natural law doctrine (see Section 3.5; cf. Welze 1962). The legal source of TJ is international law (cf. Werle and Vormann 2018, p. 27).

In a TJ process, questions of the possibilities and the limits of the *law* are at stake. As a result of globalization, conflicts are not limited to one national territory. The emergence and development of international law plays an important role in understanding the problems of the present (Hobe 2020, p. 14 ff.; Ipsen 2018, p. 21).

The definition of international law can be traced back to the *ius gentium* of Roman law (Hobe 2020, p. 8; Ipsen 2018, p. 2). According to legal scholar Stephan Hobe (2020, p. 8), *ius gentium* did not acquire an international character until the early modern period. The legal scholar Knut Ipsen (2018, p. 21) argues that constitutionalism and the history of the effects of the Enlightenment played an important role in ensuring that states were recognized as legal subjects in order to regulate state relations in international law. In connection with imperialism and colonialism, it was also explored to what extent the colonized could be regarded as having legal capacity, and whether there was an equality of people. Since there is still a discrepancy between states after decolonization and TJ mechanisms are increasingly applied in former colonial states, some countries criticize this fact as a continuation of colonialism or latent "neocolonialism" (cf. Anders and Zenker 2014, p. 396).

In the context of the TJ process in Colombia, the question therefore arises as to how Colombia itself deals with its colonial past (see Section 3.1.2; Derrida 1972, p. 427; cf. Dworkin 2014, p. 205; cf. Senn 1993, p. 73 f.). The example with land distribution shows that especially with regard to land claims there is a discrepancy between privatization and collectivization, the latter being additionally linked to a clear notion of identity (see Section 3.4.2; Jelin 1996, p. 105). Here a comparison of Derrida (1972, p. 427) and Senn (1993, p. 73 f.) can be drawn, namely with regard to the critical consideration of one's own discipline and positioning. What is meant by this is the extent to which the historical contexts are seriously and critically questioned in order to recognize them accordingly (cf. Dworkin 2014, p. 205). Accordingly, in the context of the TJ process in Colombia, it is important to keep in mind that TJ is limited in time and focused on the phase of the conflict with the former guerrilla group FARC (see Section 2.2; Werle and Vormann 2018, p. 287). Considering the time pressure and the selection of the available means, a sufficient and critical reappraisal cannot be expected. With regard to the relationship between *liberty* and *equality*, the lack of treatment remains problematic, which would, however, be necessary in order to be able to conduct a serious and critical discussion.

Finally, it should be considered that the perspective presented regarding the TJ process is predominantly transposed from a European and North American stance to a South American reality.

*3.6. Truth*

There is no *justice* without *truth.* Thus, *truth* and *justice* are important components of a functioning constitutional state. Particularly in serious cases, such as massive human rights violations within a state, the government has an obligation to ensure that what happened is investigated and brought to justice.

In a democratic state governed by the rule of law, *truth* and *justice* are elementary components alongside *liberty* and *equality*. As already mentioned in the previous chapters, *liberty, equality, democracy,* and *justice* are interpretive concepts that are related to values and are therefore subject to constant change (see Section 3.1; Dworkin 2014, p. 263). Within a constitutional state, *truth* assumes an important role because it presupposes trust and responsibility. For example, statements made by the government or government representatives should be transparent and coherent. According to Dworkin (2014, p. 24 ff.), the exercise of power should be based on political morality, which in turn is influenced by values. This is also accompanied by a responsibility, which in turn is based on moral values (Dworkin 2014, p. 31). In order to connect values and truth, an abstract interpretation is needed first. With regard to *truth* and method, Dworkin (2014, p. 305) starts from an abstract understanding of truth which, after investigation, can lead to a theory that can be applied to a concrete domain.

3.6.1. Truth and Interpretive Statements

According to Dworkin (2014, p. 36), the knowledge of *interpretive truth* depends on science and metaphysics, which cannot function without ethics. Dworkin sees respect for other people, as well as self-respect, as central, because *interpretive truth* is linked to *moral truth*. Accordingly, a critical examination of one's own science as well as one's own way of thinking is also intended, which are shaped by social, political, and cultural environments, which in each case flow into the scientific examination and, according to Dworkin (2014, pp. 22 f., 679), also into the interpretation (see Section 3.1.1; Senn 1993, p. 72 f; Gyr 2020, p. 162). For, in order to be able to discuss *truth*, according to Dworkin (2014, p. 296), *truth* must be understood as an interpretive assertion in each case. Consequently, the relation between the concept of *truth* to *truth* in everyday habitual constitutions has to be clarified in order to be able to interpret them in the next step.

Interpretation relies on analysis, which can never show the whole picture (Dworkin 2014, p. 296; cf. Gadamer 1975, p. 456). In the context of a philosophical theory of *truth*, according to Dworkin (2014, p. 296 f.), theory must always be tested against practice, for

a theory of *truth* should be able to concretely effect and justify something. According to Dworkin (2014, p. 301 f.), the theory of value in connection with the theory of moral responsibility would be suitable to come closer to this intention (see Section 3.1). Thus, it is a matter of choosing an appropriate interpretive practice that does justice to the value of the concept of *truth* (Dworkin 2014, p. 225). In the context of the TJ process in Colombia, this means that the purpose of *truth* investigation is to reveal what actually happened during the internal armed conflict in Colombia in order to prevent the recurrence of the conflict and its associated atrocities (see Section 2.2.1.; cf. SIVJRNR) In this regard, the Truth Commission has a special role in a TJ process because it was explicitly established to determine *the truth*.

### 3.6.2. Transitional Justice and Truth

*"Hay futuro si hay verdad"* (There is a future if there is truth), the slogan of the Truth Commission CEV, conveys that the future relies on truth-telling (see Section 2.2.1). A fundamental problem here, as we have seen, for example, Ángela Salazar (former member of the Truth Commission) point out in an interview, is that everyone wants *the truth*. By this she meant that different parties or interest groups claim *the truth* for themselves and thus already have a clear idea of what *the truth* should look like. Consequently, it is delicate when individuals or a group claim *the truth* for themselves, because in this way the presentation of *truth* is ultimately used as an instrument of power. Truth cannot be relativized at will in order to strengthen or enforce interests (cf. Senn 2017, p. 35). In terms of Derrida's (1972, p. 264 f.) "structurality of structure" or Dworkin's (2014, pp. 205, 269, 307) interpretive approach, *truth* is not linked to an immutable origin from which everything can be derived, since interpretive *truth* is linked to values, which in turn are subject to a dynamic process (see Section 3.1.2; cf. Senn 1993, p. 73 f.). Consequently, *truth* cannot be considered immutable. However, the more different scientific investigations are made, for example, on the same event, the better an event can be represented. A variety of methods helps to take different perspectives on the same event and expands the overall picture (cf. Gyr 2020, p. 170). However, the quality and seriousness of the investigations is crucial in determining the extent to which truth can be established in order to come to terms with the past and look to the future.

### 3.6.3. Truth and Reconciliation

In the second phase of TJ described by Teitel (2003, p. 83 f.)—the *Post-Cold War TJ*—reconciliation is the central element. Consequently, forgiveness and reconciliation become the characteristic feature of a political justification on the part of a government that—through a performative act of repentance—sets itself the goal of coming to terms with the past in order to re-enter the common future as a healed and unified state.

Wilson (2003, p. 369) also symbolizes the truth commission as the element of reconciliation. The truth commission can be understood as a constitutional and social approach to the reconciliation of conflicting parties. A truth commission comes into being through a mandate from the government and it is only established for a specific period of time (see Section 2.2; CEV 2017, Mandatos y Funciones). The task of such a commission is first and foremost that of documenting: this is how the social, historical, and structural context of the conflicts is worked through and this information is secured, in particular by producing a report documenting the human rights violations committed during the period under investigation (see Section 2.2.1). According to Wilson (2003, p. 371), following Martha Minow (2009), the report serves to bring about national reconciliation by outlining the human rights violations published in it and asking for forgiveness (cf. Gyr 2020, p. 162). In this way, the people affected should be able to come to terms with the past and their conflicts, as the slogan of the CEV also points out (see Section 2.2.1). In this sense, a line should also be drawn so that the government can start anew in order to come to terms with the past and can no longer be held legally responsible (Wilson 2003, p. 371; cf. Gyr 2020, p. 158).

According to Wilson (2003, p. 371 f.), the documentation of human rights violations must therefore be considered in the context of social scientific and jurisprudential positivism, as it has become the accepted method in relation to the search for *truth* in TJ. A positivist understanding of *truth* is determined by tangible factors, which should be as value-free as possible and thus dispense with *epistemological reflection* (cf. Gyr 2020, p. 159). Accordingly, a positivist understanding of *truth* alone is not sufficient to satisfy Dworkin's (2014, p. 301 f.) requirements with regard to *truth* as an interpretive assertion (see Section 3.6.1). For example, according to the value theory, moral responsibility would have to be claimed with respect to the understanding of truth, which is not the case in positivism (see Section 3.1). Consequently, a positivist understanding of *truth* cannot suffice to bring about reconciliation. In the end, no method alone is sufficient to adequately portray the complex interrelationships of an internal armed conflict that has lasted for decades within such a short period of time.

## 4. Conclusions

According to Roland Dworkin's (2011, 2014) interpretive theory, the relationship between *liberty* and *justice* in relation to the search for *truth* within a TJ process is linked to value concepts. Values, Dworkin (2014, p. 29) argues, while independent in themselves, are also linked to other values, which thereby support each other. According to Dworkin (2014, pp. 13, 207), truth occupies a special position: on the one hand, because there is disagreement about what truth is in the first place, and on the other hand, because truth in connection with other values makes a successful way of life possible (see Section 3). According to Dworkin (2014, pp. 22 f., 678 f.), a successful way of life is achievable in a liberal constitutional state (see Section 3; cf. Ibric 2022, p. 139). On the one hand, citizens were able to co-determine these legal foundations, and on the other hand, these legal foundations are equally binding for all. *Law* and *morality* are thus intertwined and remain interpretive. Accordingly, the analysis of the terms goes hand in hand with the identification of political, economic, and social practices (Dworkin 2014, pp. 22 f., 678 f.).

In a TJ process, the peaceful coexistence of a society is to be made possible again through the determination of *truth* and *justice,* so that the basic requirements of a constitutional state are also restored (see Section 2.1; Teitel 2003, p. 69 f.; Werle and Vormann 2018; cf. Gyr 2020, p. 156 f.).

The TJ process in Colombia provides an example of how a complex and protracted internal armed conflict can significantly damage trust between the government and the population on the one hand, and the various strata within the population on the other.

Roland Dworkin's (2014, pp. 263 fff., 269, 307) theory of justice is built on the foundation of a liberal rule of law, which in practice is characterized by active and constant interpretation (see Section 3). It is basically the process that arises by itself within a state-constituted society because people always deal with key issues as well as with everyday questions. Therefore, *liberty* and *justice* in relation to finding the *truth* in a TJ process are also concepts that always have to be discussed from the concrete social, cultural, economic, and political conditions of a society (cf. Dworkin 2014, pp. 205, 224, 269, 307; cf. Derrida 1972, p. 423; cf. Senn 1993, p. 73 f.) But according to Dworkin (2014, pp. 172 ff., 222, 268) such an interpretation is always connected with a personal moral responsibility, which can be compared with the *scientific self-reflection* according to Senn (1993, p. 72 f.) and the *structurality of structure* according to Derrida (1972, p. 427).

Thus, a critical examination of one's own discipline as well as one's own attitude is indispensable. Therefore, moral responsibility is also to be judged interpretatively and on the basis of a further moral interpretation (see Section 4, Dworkin 2014, p. 174). An interpretation of *justice* is thus always accompanied by another moral interpretation, which in turn provokes a questioning, which provokes an ongoing discussion on the concept (see Section 3; Dworkin 2014, pp. 274, 601 f.). This circular procedure, however, represents every discourse in science and in a political discussion as a "normal procedure". In this

sense, various reference parameters are always considered and (usually only provisionally) evaluated.

Based on Dworkin's interpretive theory, the greatest challenge in a TJ process is to clarify and overcome the disagreement over fundamental values that have led to internal armed conflict. The unequal distribution of power, land rights and resources has usually been the triggering event for armed guerrillas to be formed (see Section 3.3.2; Farnsworth-Alvear et al. 2017, p. 343). This is particularly true in Colombia. The conflict, which has lasted for more than fifty years, has led to the formation of other parties to the conflict, such as drug cartels (see Section 3.3.2; Richani 1997, p. 37; Werle and Vormann 2018, p. 286). Countless massive human rights violations committed by the parties to the conflict, including the government, rob the rule of law of its legitimacy to guarantee *liberty* and *equality* through *law* (see Section 3.4; Senn 2022, p. 3; cf. Habermas 1992, with regard to "social integration", p. 527; cf. Mahlmann 2019, with regard to "Functions of the law", pp. 32–37).

This situation is consequently to be rectified by a TJ process.

Through its process, TJ seeks to create new stable conditions leading to a liberal rule of law (see chapter 2 with regard to Teitel 2003). Within the mechanisms of SIVJRNR, the Special Court JEP is the most important. The mandate of the Truth Commission CEV is usually short in relation to the time span involved in the internal armed conflict. Due to the limited time span of the mechanisms as well as the ambitious goal of re-establishing a just state, all participants are under high pressure to succeed. However, the overly rigorous demand for results to be available as quickly as possible does not encourage a more in-depth reflective examination of the subject under study and how it should be appropriately presented afterwards so that it is not re-historicized (cf. Senn 1993, p. 73 f.; cf. Gyr 2020, p. 161). In that sense, it must be possible to bring in the interpretive approach, as Dworkin (2011, 2014) points out, in order to be able to have a "normal" discussion, which cannot be concluded hastily and thus will not provide a reliable basis for the new society. The reason for this is that TJ represents a constant process and change that is shaped by global and regional events and hence cannot be explained from only one perspective.

**Funding:** This research received no external funding.

**Conflicts of Interest:** The author declares no conflict of interest.

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
