# Peer review of "Transitional Justice Process and the Justice Theory of Roland Dworkin"

_laws_

Round 1
Reviewer 1 Report
Very interesting and original article.
I suggest to the author in the introduction to explain deeply the concept of TJ, pointing out also the relevance for other Latin American experiences (for example Brazil) and to problematize the concept. At least to mention some the Latin America experiences.
Author Response
Thank you very much for your feedback. I have taken your comments into account and added them to the introduction accordingly. At the same time, I have carefully reread the entire text and made linguistic improvements.
Best regards

Reviewer 2 Report
In my view the study is about the question to what extent some theses of Dworkin's philosophy of law about truth and justice can be made fruitful when a society wants/needs to move back under the rule of law after a prolonged period of the rule of power and discord. This necessarily requires that all those involved agree on an understanding of the truth about what has happened. Only on the basis of truth, which has been worked out in a discursive process between different points of view, can justice be realised. The author uses the example of truth processing in Colombia, which fits well here. The topic of truth commissions will continue to be of great relevance in the future, as many cases of application are already emerging today. In many states, rule is not subject to law and is characterised by power struggles. Law appears as a mere instrument of rule and appears arbitrary. Such phases of the dissolution of law leave deep scars that can only be healed by a careful reappraisal of the question of truth. There is a lot of literature on the subject. The submitted contribution is, in my estimation, original and fills a gap. However, I cannot discuss this in detail because I am only rudimentarily familiar with the specific published material. A specialised expert might have to be consulted. There is no specific method in the philosophy of law as there is in the natural and social sciences. It is essential that the study is coherent in itself and that the results as a whole are logically convincing. In my opinion, this is the case here. The conclusions are consistent with the arguments presented. They are related to the objective of the paper. The reference to literature and the references are appropriate. The appreciation of such a contribution, which puts abstract philosophical treatises to the test with concrete cases of application, can vary. I like the contribution. It is innovative and can certainly be discussed controversially. I therefore recommend its inclusion in the anthology.
Author Response
Thank you very much for your feedback. I have carefully reread the entire text and made linguistic improvements.
Best regards
